# Use of the MNCD Classification to Monitor Clinical Stage and Response to Levodopa-Entacapone-Carbidopa Intestinal Gel Infusion in Advanced Parkinson’s Disease

**DOI:** 10.3390/brainsci14121244

**Published:** 2024-12-12

**Authors:** Diego Santos-García, Lydia López-Manzanares, Inés Muro, Pablo Lorenzo-Barreto, Elena Casas Peña, Rocío García-Ramos, Tamara Fernández Valle, Carlos Morata-Martínez, Raquel Baviera-Muñoz, Irene Martínez-Torres, María Álvarez-Sauco, Déborah Alonso-Modino, Inés Legarda, María Fuensanta Valero-García, José Andrés Suárez-Muñoz, Juan Carlos Martínez-Castrillo, Ana Belén Perona, Jose María Salom, Esther Cubo, Caridad Valero-Merino, Nuria López-Ariztegui, Pilar Sánchez Alonso, Sabela Novo Ponte, Elisa Gamo Gónzález, Raquel Martín García, Raúl Espinosa, Mar Carmona, Cici Esmerali Feliz, Pedro García Ruíz, Teresa Muñoz Ruíz, Beatriz Fernández Rodríguez, Marina Mata Alvarez-Santullano

**Affiliations:** 1Department of Neurology, Hospital Universitario de A Coruña (HUAC), Complejo Hospitalario Universitario de A Coruña (CHUAC), C/As Xubias 84, 15006 A Coruña, Spain; 2Grupo de Investigación en Enfermedad de Parkinson y otros Trastornos del Movimiento, INIBIC (Instituto de Investigación Biomédica de A Coruña), 15006 A Coruña, Spain; 3Hospital San Rafael, 15006 A Coruña, Spain; 4Fundación Degen, 15006 A Coruña, Spain; 5Hospital Universitario La Princesa, 28006 Madrid, Spain; lydia.lopez@salud.madrid.org (L.L.-M.); ines.murog@gmail.com (I.M.); pablorenzobarreto@gmail.com (P.L.-B.); ecasasp@salud.madrid.org (E.C.P.); 6Hospital Clínico Universitario San Carlos, 28040 Madrid, Spain; rocio.garciaramos@salud.madrid.org; 7Hospital de Cruces, 48903 Barakaldo, Spain; tamara.fernandezvalle@osakidetza.eus; 8Hospital Universitario la Fe, 46026 Valencia, Spain; morata_carmara@gva.es (C.M.-M.); raquelbaviera@gmail.com (R.B.-M.); irenemto@hotmail.com (I.M.-T.); 9Hospital General Universitario de Elche, 03203 Elche, Spain; mariaalsa@hotmail.com; 10Hospital Universitario de la Candelaria, 38010 Santa Cruz de Tenerife, Spain; dalomod@gobiernodecanarias.org; 11Hospital Universitario Son Espases, 07120 Palma de Mallorca, Spain; ines.legarda@ssib.es (I.L.); mf.v.g23@gmail.com (M.F.V.-G.); 12Hospital Dr. Negrín, 35010 Las Palmas de Gran Canaria, Spain; ppsumu@gmail.com; 13Hospital Universitario Ramón y Cajal, 28034 Madrid, Spain; jmcastrillo@salud.madrid.org; 14Complejo Hospitalario Universitario de Albacete, 02006 Albacete, Spain; abperona@sescam.jccm.es; 15Hospital Clínico Universitario de Valencia, 46010 Valencia, Spain; jomasaju@hotmail.es; 16Hospital Universitario de Burgos, 09006 Burgos, Spain; esthercubo@gmail.com; 17Hospital Arnau de Vilanova, 46015 Valencia, Spain; caridadvaleromerino@gmail.com; 18Hospital Universitario de Toledo, 45007 Toledo, Spain; nlopeza@sescam.jccm.es; 19Hospital Puerta de Hierro, Majadahonda, 28222 Madrid, Spain; msalonso@salud.madrid.org (P.S.A.); snovoponte@gmail.com (S.N.P.); eligg91@hotmail.com (E.G.G.); raquel.marting92@gmail.com (R.M.G.); 20Hospital Universitario de Jerez, 11407 Jerez, Spain; raulespinosarosso@gmail.com; 21Hospital Universitario de Basurto, 48013 Bilbao, Spain; mariadelmar.carmonaabellan@osakidetza.eus; 22Hospital Fundación Jiménez Díaz, 28040 Madrid, Spain; cefeliz@quironsalud.es (C.E.F.); pgarcia@fjd.es (P.G.R.); 23Hospital Regional Universitario de Málaga, 29010 Málaga, Spain; munozruiz.teresa@gmail.com (T.M.R.); beabeafdez@gmail.com (B.F.R.); 24Hospital Infanta Sofía, San Sebastián de los Reyes, 28702 Madrid, Spain; mmataal@yahoo.es

**Keywords:** advanced, device-aided therapy, MNCD, non-motor symptoms, parkinson’s disease

## Abstract

Background and objective: Staging Parkinson’s disease (PD) with a novel simple classification called MNCD, based on four axes (Motor; Non-motor; Cognition; Dependency) and five stages, correlated with disease severity, patients’ quality of life and caregivers’ strain and burden. Our aim was to apply the MNCD classification in advanced PD patients treated with device-aided therapy (DAT). Patients and Methods: A multicenter observational retrospective study of the first patients to start the levodopa-entacapone-carbidopa intestinal gel (LECIG) in Spain was performed (LECIPARK study). The MNCD total score (from 0 to 12) and MNCD stages (from 1 to 5) were collected by the neurologist at V0 (before starting LECIG) and V2 (follow-up visit). Wilcoxon’s signed rank and Marginal Homogeneity tests were applied to compare changes from V0 to V2. Results: Sixty-seven PD patients (58.2% males; 69.9 ± 9.3 years old) with a mean disease duration of 14.4 ± 6.5 years were included. The mean treatment duration (V2) was 172.9 ± 105.2 days. At V0, patients were classified as in stage 2 (35.8%), 3 (46.3%) or 4 (17.9%). The frequency of patients in stage 4 decreased to 9% at V2 (*p* = 0.001). The MNCD total score decreased from 6.27 ± 1.94 at V0 to 5.21 ± 2.23 (*p* < 0.0001). From V0 to V2, the motor (M; *p* < 0.0001) and non-motor symptom (N; *p* < 0.0001) burden decreased, and autonomy for the activities of daily living (D; *p* = 0.005) improved. Conclusions: The MNCD classification could be useful to classify advanced PD patients and to monitor the response to a DAT.

## 1. Introduction

Parkinson’s disease (PD) is a complex and very heterogeneous progressive neurodegenerative disorder causing not only motor but also non-motor symptoms (NMS) that result in loss of patient autonomy for activities of daily living (ADL) and a worse quality of life (QoL) [1,2,3]. From a clinical point of view and given the great variability of outcomes in PD, it is essential to have an easy-to-use tool that allows the staging of PD. Recently, a new classification called “MNCD” has been proposed [4]. The MNCD is based on four major axes (M, Motor; N, Non-motor; C, Cognition; D, Dependency) and proposes five stages (MNCDst), from MNCD stage 1, no relevant symptoms, to MNCD stage 5, dementia and dependency for basic activities of daily living (ADL), and a total score (MNCDsc) from 0 (0 + 0 + 0 + 0 = 0; the best possible status) to 12 (4 + 4 + 2 + 2 = 12; the worst possible status). We demonstrated using data from the Spanish PD cohort COPPADIS [5,6] that PD staging applying the MNCD classification correlated with disease severity, patients’ QoL [7], and caregiver’ burden [8]. Moreover, a group from China observed in a cohort of 357 PD patients that the correlation of the MNCD staging with the QoL was more statistically significant than to the Hoehn&Yahr staging [9]. Originally, the MNCD was proposed as a tool to monitor the progression of PD, from the first moment (at diagnosis) to the end of the follow-up of the patient. It could be even used in cohort studies or clinical trials, especially in those with a long follow-up (e.g., disease-modifying therapies) [10]. Interestingly, the MNCD stage could change from a higher to a lower stage after treatment in some cases (e.g., the third example case in the original description of the classification [4]). Regarding this aspect, we suggested that the MNCD could be useful to be applied in advanced PD patients treated with a device-aided therapy (DAT), both to classify the patient according to all information collected with the MNCD but also to monitor the response to the DAT.

The aim of this study was to apply, for the first time, the MNCD classification in advanced PD patients treated with a DAT. Specifically, the change in the characteristics, MNCDst and the MNCDsc from before to after levodopa-entacapone-carbidopa intestinal gel (LECIG) infusion was analyzed in advanced PD patients from the very recently reported LECIPARK study [11].

## 2. Material and Methods

A multicenter, longitudinal, retrospective, observational study of the first patients to start LECIG in Spain was performed (LECIPARK [descriptive analysis about the use of LECIgon in patients with PARKnson’s disease in Spain]) [11]. All centers from Spain with an experience of at least 2 PD patients treated with LECIG until 31 March 2024, were invited to participate. The data were collected from three different time points: V0, an indication of therapy (LECIG) by the neurologist; V1, initiation of LECIG; V2, a follow-up visit. The data for visits V0 and V1 were collected from the medical records whereas the data for visit V2 were collected from a specific data report registry assessed in the clinic. The period for collecting the data was 6 months, from December 2023 to May 2024. Information on sociodemographic aspects, comorbidity, factors related to PD, and treatment including LECIG and the levodopa equivalent daily dose (LEDD) [12] was collected [11].

The MNCD classification was applied by the neurologist at V0 and at V2. This classification [4] is based on four axes: (1) Motor symptoms; (2) Non-motor symptoms; (3) Cognition; and (4) Dependency for ADL. The first axis (Motor symptoms) is subdivided into four defined sub axes: (1) motor fluctuations; (2) dyskinesia; (3) axial symptoms; and (4) tremor. The second axis (Non-motor symptoms) is subdivided into four defined sub axes: (1) neuropsychiatric symptoms; (2) autonomic dysfunction; (3) sleep disturbances and fatigue; and (4) pain and sensory disorders. Regarding the third axis (Cognition), patients are classified as having normal cognition, with mild cognitive impairment or dementia. Finally, patients are classified according to the fourth axis (Dependency) as having independence for activities of daily living, with dependency for instrumental or with dependency for basic activities. Patients were classified into five groups according to the MNCDst: Stage 1 (the patient has no relevant symptoms); Stage 2 (there is at least 1 motor symptom or 1 NMS scoring in the MNCD classification); Stage 3 (there is mild cognitive impairment and/or dependency for instrumental ADL); Stage 4 (there is dependency for basic ADL but no dementia); and Stage 5 (there is dementia and dependency for ADL). Moreover, the MNCDsc (from 0 to 12) was calculated according to the sum of the score of all axes of the MNCD classification: M (from 0 to 4) + N (from 0 to 4) + C (from 0 to 2) + D (from 0 to 2).

### 2.1. Statistical Analysis

Data were processed using SPSS 20.0 for Windows. Continuous variables were presented as mean  ±  standard deviation (SD) or median (interquartile range), while categorical variables were expressed as n (%). The distribution for variables was verified by a one-sample Kolmogorov–Smirnov test. Wilcoxon’s signed rank and Marginal Homogeneity tests were applied to compare changes from V0 to V2. Spearman’s or Pearson’s correlation coefficient, as appropriate, was used to analyze the relationship between the change in both the MNCDst and MNCDsc and the clinical global impression of change (CGI-C; 1, very much worse; 2, much worse; 3, minimally worse; 4, no change; 5, minimally improved; 6, much improved; 7, very much improved) and visual analogue scale global improvement (VAS-GI; from 0, the worst, to 10, the best improvement) according to the opinion of the neurologist, patient and principal caregiver. Correlations were considered weak for coefficient values ≤ 0.29, moderate for values between 0.30 and 0.59, and strong for values ≥ 0.60. The value of *p* was considered significant when it was <0.05.

### 2.2. Standard Protocol Approvals, Registrations, and Patient Consents

For this study, we received approval from the Comité de Ética de Investigación de medicamentos de Galicia (CEImG) from Spain (2023/527; 19 November 2023). Written informed consents from all participants in this study was obtained.

### 2.3. Data Availability

The protocol, statistical analysis plan and data are available on request.

## 3. Results

A total of 67 out of 73 PD patients (58.2% males; 69.9 ± 9.3 years old) from the LECIPARK study were included in the analysis (91.8% of the sample; 6 patients without data collected about the MNCD). The mean disease duration at baseline was 14.4 ± 6.5 years (range, 5–31). At V0, the mean OFF time (N = 65) was 5.2 ± 3 h (range, 1–15) and 74.6% and 80.3% of the patients had non-motor fluctuations and dyskinesia, respectively. Other characteristics related to PD and treatment at the baseline are shown in Table 1. The mean exposure to LECIG was 172.9 ± 105.2 days (range, 7–476). The follow-up time was ≥3 months and ≥6 months in 76.2% and 47.6% of the patients, respectively.

The results represent %, mean ± SD or median [p25, p75]. BMI, body mass index; COMT, catechol-O-methyl transferase; DA, dopamine agonist; DAT, device-aided therapy; DBS, deep brain stimulation; H&Y, Hoehn&Yahr; LCIG, levodopa-carbidopa infusion gel; LECGI, levodopa-entacapone-carbidopa infusion gel; LEDD, levodopa equivalent daily dose; MAO, monoamine oxidase; MCI, mild cognitive impairment; UPDRS, Unified Parkinson’s Disease Rating Scale.

At V0, patients were classified as in stage 2 (35.8%), 3 (46.3%) or 4 (17.9%). From V0 to V2, a significant change was detected in the MNCDst with a decrease in the percentage of stage 4 (from 17.9% to 9%) and an increase in stage 2 (from 35.8% to 44.8%) (*p* = 0.001) (Figure 1A). Two patients were even classified as in MNCD stage 1 at V2. The number of patients with a greater score (from 0 to 4) decreased significantly from V0 to V2 regarding motor symptoms (M; *p* < 0.0001) and NMS (N; *p* < 0.0001) (Figure 1B). Independence for ADL was observed in 33 patients at V2 compared to 27 at baseline (*p* = 0.005) (Figure 1B). No differences were detected in cognition (*p* = 1.000). Regarding the MNCDsc, a significant decrease from V0 to V2 was detected in the total (from 6.27 ± 1.94 at V0 to 5.21 ± 2.23; *p* < 0.0001), M (from 2.73 ± 0.71 at V0 to 2.24 ± 0.8; *p* < 0.0001), N (from 2.51 ± 1.01 at V0 to 2.1 ± 1.11; *p* < 0.0001) and D (from 0.78 ± 0.73 at V0 to 0.59 ± 0.65; *p* = 0.005) scores (Figure 2). No correlation was found between the change from V0 to V2 in the MNCDst and the CGI-C or VAS-GI but there was between the change in the MNCDsc and the CGI-C and the MNCDsc and the VAS-GI (Table 2). Regarding the motor stage, no significant changes were detected in this sample (N = 67) in the H&Y during the ON state from before (2 [2, 3]); 2.5 ± 0.6) to after (2 [2, 3]); 2.2 ± 0.7) LECIG (*p* = 0.649).

## 4. Discussion

The MNCD is a new proposed classification for PD based on four axes [4] that has been demonstrated to correlate with disease severity, patients’ QoL, and caregivers’ strain and burden [7,8,9]. It has been designed to monitor the progression of PD and hypothetically, even the response to a therapy [4]. Here, we applied, for the first time, the MNCD in patients with advanced PD treated with a DAT. Very interestingly, a significant improvement in the MNCD stage and the MNCD score was detected in 67 PD patients treated with LECIG infusion therapy. Moreover, the improvement in the MNCD score correlated with the clinical global impression of change (improvement) according to the neurologist, the patient, and the principal caregiver’s opinion. From a clinical point of view and based on these findings, we suggest that the MNCD classification could be useful to apply in advanced PD patients in daily clinical practical.

A critical factor selecting a patient for a DAT is the proper indication of the therapy. Differences in access to care, referral pattern (timing and frequency), as well as physician biases (unconscious/implicit or conscious/explicit bias) and patients’ preferences or health-seeking behavior are to be considered [13]. However, a critical factor is the necessity to conduct a very complete evaluation collecting a lot of information about different aspect of the disease and other comorbidities [14]. In this context, some tools have been developed with the aim of helping the physician to select an advanced PD patient for a DAT such as the CDEPA questionnaire, 5-2-1 criteria, MANAGE-PD, D-DATS, FLASQ-PD, and Stimulus 1 or Stimulus 2 [15]. In a recent narrative review, Moes et al. [15] briefly included the “MNCD tool” in the “Other screening tools and testing methods” section. Although the MNCD classification was not specifically designed to apply in advanced PD patients as a tool to help in the selection of a DAT as Moes et al. commented, our novel findings can be interpreted as it could be useful due to the amount of information with this tool being collected (motor fluctuations; dyskinesias; axial symptoms; refractory tremor; neuropsychiatric symptoms; sleep/fatigue; dysautonomic symptoms; pain and sensory symptoms; cognitive status; disability or dependency for ADL). Importantly, the classification shows the information in such a way that it can be visually interpreted very quickly and the stage and score can also help to classify the patient as more advanced or with a greater burden of symptoms that generate disability. As expected, of 67 advanced PD patients selected to be treated with LECIG, there were no cases with a MNCD stage 5 (dementia and dependency for basic ADL) due to the fact that a patient in stage 5 would not be a good candidate for a DAT. On the other side, only one out of three patients were advanced PD patients with normal cognition and independence for instrumental and basic ADL. Moreover, only two patients at baseline had no relevant NMS (N0) whereas nearly one out five patients had at least one relevant NMS of each sub-axis (N4).

A very interesting finding is that the improvement obtained by patients with LECIG, recently reported [11], was reflected in the MNCD classification, both in stage and score. After starting with LECIG, the number of patients in stage 4 was reduced to half (from 12 to 6), and 2 patients were even classified as stage 1 (without relevant motor and NMS). Moreover, the motor and non-motor burden decreased as it was reflected in the total, M and N MNCD scores, and the number of patients reporting symptoms (M and N). In example 3 of the original description of the MNCD classification [4], we suggested that a patient could hypothetically pass from a higher to a lower MNCD stage and decrease the symptoms burden and global score after effective specific treatment (e.g., refractory tremor improvement after ultrasound therapy and impulse control disorder remission after dopamine agonist withdrawal passing from stage 2 to 1 and from score 2 [0001/1000/0/0] to 0 [0000/0000/0/0]). Here, in advanced PD patients, a critical factor explaining the changes in the MNCDst was the improvement in disability, again reflected in the score (D score). Curiously, although gaining autonomy and independence is one of the objectives of treating symptomatically with a DAT, it is an aspect that is often not properly evaluated in advanced PD patients in favor of others such as QoL or the OFF-time reduction [16]. The MNCD classification is simple but could be used to monitor the response to a DAT (motor symptoms; NMS; cognition; dependency) not only in the short- (i.e., a possible improvement in some cases reflected in the MNCDst and MNCDsc due to the DAT effect) but also in the long-term (i.e., impairment reflected in the MNCDst and MNCDsc due to the progression of the disease). Finally, a correlation was observed between the improvement in the MNCDsc and the improvement perception reported by the neurologist, the patient, and the principal caregiver. This agrees with known improvement very frequently reported by patients and physicians after being treated with a DAT [17,18].

The present study has some limitations. Firstly, limitations related to the retrospective observational design. However, much information was available and collected from the medical records due to patients being exhaustively evaluated in centers with experience from Spain in the management of DATs. Secondly, only patients treated with LECIG were included in this study but not with other DATs. Thirdly, the sample was small (N = 67) and the mean follow-up was short (mean of about 6 months), making it necessary to conduct a study applying the MNCD in a big cohort of PD patients treated with different DATs with a short- and long-term follow-up. Fourthly, the changes in the MNCD classification were not directly compared to other scales. However, significant changes were detected in M, N and D scores and also in the MNCD total score from pre to post-LECIG but not in the H&Y during the ON state. More information is collected with the MNCD and it is also more sensible to change the H&Y. Fifthly, non-parametric tests were applied to analyze the change in different variables from V0 to V2 according to the observational design, so the influence of covariates was not taken into consideration. On the other hand, this is the first time that the MNCD classification is used in advanced PD patients to classify and monitor the response to a DAT.

## 5. Conclusions

In conclusion, the MNCD classification was applied here for the first time in advanced PD patients treated with a DAT. Our findings suggest that it could be useful to classify advanced PD patients and to monitor the response to a DAT. More data are needed to know the possible role of the MNCD classification as a tool to use to monitor PD progression.

## Figures and Tables

**Figure 1 brainsci-14-01244-f001:**
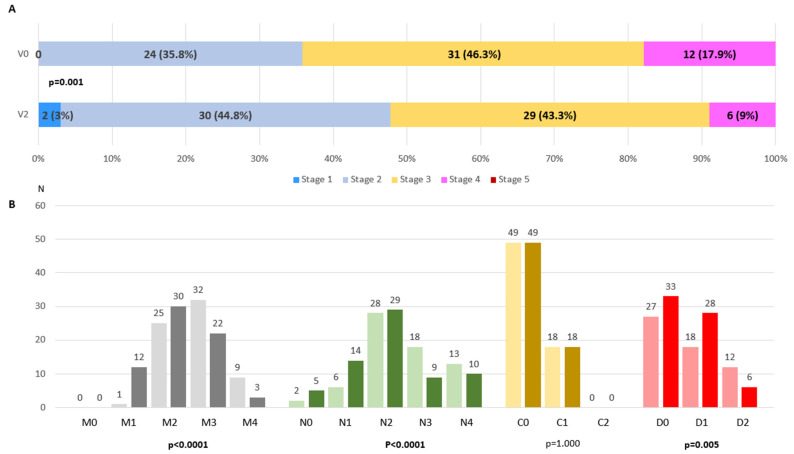
(**A**). Frequency of different MNCD stages (from 1 to 5) at V0 (V0; pre-LECIG) compared to at V2 (follow-up visit; 172.9 ± 105.2 days after starting LECIG) (*p* < 0.0001). Marginal Homogeneity test applied. (**B**). Number of patients with each score of the MNCD classification (M, N, C, and D) at V0 (on the left for each score, in light color) compared to at V2 (on the right for each score, in dark color). Marginal Homogeneity test applied.

**Figure 2 brainsci-14-01244-f002:**
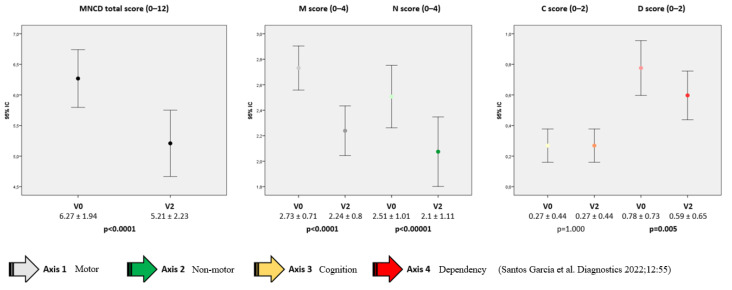
Change from the baseline (V0; pre-LECIG) to the follow-up visit (V2) in the MNCD total score (from 0 to 12), M and N scores (from 0 to 4), and C and D scores (from 0 to 2). Wilcoxon’s signed rank tests were applied. The bars represent mean ± standard deviation.

**Table 1 brainsci-14-01244-t001:** Data about sociodemographic aspects, comorbidities, PD, antiparkinsonian drugs and other therapies at baseline (V0).

	N			N	
Age	67	69.9 ± 9.3 (42–85)	Time with fluctuations (years)	64	7.2 ± 4.2 (2–20)
Gender (males) (%)	67	58.2	Non-motor fluctuations (%)	67	74.6
			Daily OFF time (hours)	65	5.3 ± 3 (1–15)
Weight (kg)	50	67.9 ± 12.3 (47–102)	H&Y–OFF	67	3 [3, 4]
Height (cms)	51	165 ± 9.4 (142–185)	H&Y–ON	67	2 [2, 3]
BMI	49	25.2 ± 3.9 (18.9–36)	UPDRS–III–OFF	58	42.8 ± 16.6 (26–78)
			UPDRS–III–ON	60	20.9 ± 11.2 (0–40)
Civil status (%):	58		Dyskinesia (%)	61	80.3
- Married		63.8			
- Single		17.2	Entacapone previously (%)	67	63
- Widowed		8.6	DAT previously (%):	67	
- Other		10.4	- DBS		4.5
			- Apomorphine		7.5
Living style (%):	63		- LCIG		22.4
- With the partner		58.7	- More than 1		11.9
- Alone		11.1	- Other		1.5
- With another family member		7.9			
- Other		22.3	Treatment for PD (%):	67	
			- Levodopa		100
Comorbidities (%):	67		- LCIG		34.3
- Arterial hypertension		28.4	- MAO-B inhibitor		56.7
- Diabetes mellitus		14.9	- Dopamine agonist		53.7
- Dyslipemia		26.9	- COMT inhibitor		50.7
- Atrial fibrillation		9	* Entacapone		25.4
- Cardiopathy		7.5	* Opicapone		25.4
- Lung disease		3	- Amantadine		29.9
- Polineuropathy		7.5			
			L-dopa daily dose (mg)	61	1078.2 ± 464.9 (400–2.448)
Time from diagnosis (years)	67	14.4 ± 6.5 (5–31)	DA daily dose (mg)	35	235.7 ± 214.9 (26–880)
Motor phenotype (%):	66		LEDD (mg)	60	1485.3 ± 500.4 (500–2.660)
- Tremor dominant		28.8			
- Indeterminate		34.8	Other treatments (%):	67	
- PIGD		36.4	- Antidepressant		53.7
Cognitive impairment (%):	67		- Benzodiazepine		49.3
- MCI (%)		26.9	- Antipsychotic		22.4
- Dementia (%)		1.5	- Anti-dementia		11.9

**Table 2 brainsci-14-01244-t002:** Correlation (r) between the change from V0 to V2 (Δ = value at V0–value at V2) and the CGI-C and the VAS-GI.

	Δ MNCD Stage	Δ MNCD Score
CGI-C Neurologist	0.036 (N = 65; *p* = 0.774)	0.370 (N = 65; *p* = 0.002)
CGI-C Patient	−0.001 (N = 67; *p* = 0.992)	0.355 (N = 67; *p* = 0.003)
CGI-C Principal caregiver	0.209 (N = 61; *p* = 0.106)	0.474 (N = 61; *p* < 0.0001)
VAS-GI Neurologist	0.060 (N = 65; *p* = 0.635)	0.338 (N = 65; *p* = 0.006)
VAS-GI Patient	0.069 (N = 65; *p* = 0.585)	0.315 (N = 65; *p* = 0.011)
VAS-GI Principal caregiver	0.178 (N = 60; *p* = 0.175)	0.351 (N = 60; *p* = 0.006)

The results represent Spearman’s correlation coefficient. CGI-C, Clinical Global Impression of Change (from 1, very much worse, to 7, very much improved); VAS, Visual Analog Scale Global Improvement (from 0, the worst, to 10, the best improvement).

## Data Availability

The protocol, statistical analysis plan, and data are available on request. This can be requested from the promoter of the study, Diego Santos García, by sending an email to the following address: diegosangar@yahoo.es.

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
