# Peer review of "Use of the MNCD Classification to Monitor Clinical Stage and Response to Levodopa-Entacapone-Carbidopa Intestinal Gel Infusion in Advanced Parkinson’s Disease"

_brainsci, 2024, doi:10.3390/brainsci14121244_

Round 1

Reviewer 1 Report

Comments and Suggestions for Authors

The authors have looked at the use of the MNCD scale for clinical staging and treatment response in individuals with Parkinson's disease. Although I appreciate the usefulness of exploring other ways of looking at disease progression in PD and looking beyond the classical and probably outdated scales such as Hoehn and Yahr staging, I have a few concerns and comments in relation to the study. My main concern is whether with the current design of the study it is possible to comment on the reliability of the MNCD scale to measure disease staging and progression. 

1. The title is slightly misleading in my opinion as the use of the MNCD scale is limited to individuals with PD in an advanced stage and being started on a specific device-aided therapy. I would suggest clarifying this in the title.

2. Do the authors feel they might have introduced a selection bias by only including individuals with PD being considered for Lecig? 

3. Why was data for MNCD scale outcomes missing for 6 participants? In this respect, it would be useful to clarify the standard protocol for follow-up of people initiated on Lecig or device-aided therapies in general. 

4. Why did the authors not compare the changes in MNCD scores to other standard outcomes, such as the UPDRS or Hoehn and Yahr staging to understand the behaviour of the new scale compared to the ''standard'' assessment. 

5. I noticed that almost 50% of participants in this study have had previous device-aided therapies. Do the authors feel this may have influenced their outcomes? 

6. At what time point did visits take place? Was, for example, V2 a standard point in time or was this based on when a participant was seen in clinic and, therefore, possibly random? Understanding the timelines of the visits would be crucial. 

7. The text in many of the figures is small and difficult to read. 

Author Response

The authors have looked at the use of the MNCD scale for clinical staging and treatment response in individuals with Parkinson's disease. Although I appreciate the usefulness of exploring other ways of looking at disease progression in PD and looking beyond the classical and probably outdated scales such as Hoehn and Yahr staging, I have a few concerns and comments in relation to the study. My main concern is whether with the current design of the study it is possible to comment on the reliability of the MNCD scale to measure disease staging and progression. 

  1. The title is slightly misleading in my opinion as the use of the MNCD scale is limited to individuals with PD in an advanced stage and being started on a specific device-aided therapy. I would suggest clarifying this in the title.

Many thanks for your comment. We agree with you. The title has changed to “Use of the MNCD Classification to Monitor Clinical Stage and Response to Levodopa-entacapone-carbidopa Intestinal Gel Infusion in Advanced Parkinson´s Disease”.

  1. Do the authors feel they might have introduced a selection bias by only including individuals with PD being considered for Lecig? 

Many thanks for your comment. This analysis is a part of the planned LECIPARK study. That it has been already executed. We agree with you that it would be necessary to analyze prospectively the change in the MNCD for all patients with advanced PD treated with a DAT (i.e., DBS and infusion therapy). This aspect has been included in the Discussion: “Secondly, only patients treated with LECIG were included in this study but no with other DATs. Thirdly, the sample was small (N=67) and the mean follow-up short (mean about 6 months), making it necessary to conduct a study applying the MNCD in a big cohort of PD patients treated with different DATs with a short- and long-term follow-up”.

  1. Why was data for MNCD scale outcomes missing for 6 participants? In this respect, it would be useful to clarify the standard protocol for follow-up of people initiated on Lecig or device-aided therapies in general. 

Many thanks for your comment. Six out of 73 is a low percentage (8.2%). The MNCD was applied at V0 and V2. The reason is that the neurologist hadn´t collect this information at V0 (pre-LECIG). Although all patients were treated by expert neurologists on advanced PD from Spanish Movement Disorders Units and exhaustive evaluations are conducted in daily clinical practice, no all data are always collected. This is a retrospective study and of course, as it has been commented, this is an important limitation and the explanation why the N is not 73 for all subjects from the LECIPARK study (please, see Santos García et al. Eur J Neurol 2024 [epub ahead of print], reference 11).

  1. Why did the authors not compare the changes in MNCD scores to other standard outcomes, such as the UPDRS or Hoehn and Yahr staging to understand the behaviour of the new scale compared to the ''standard'' assessment. 

Many thanks for your comment. The aim of this study was to describe in this cohort the results of applying the MNCD before and after been treated with LECIG. But not to compare with another scale. As we commented in the Introduction, a group from China observed in a cohort of 357 PD patients that the correlation of the MNCD staging with the QoL was more statistically significant than to the Hoehn&Yahr staging [reference 9]. The UPDRS-III and H&Y were assessed during the OFF and during the ON state before but only during the ON state after receiving LECIG (patients go to the clinic in daily clinical practice in the ON state when they are receiving levodopa infusion). No significant changes were detected in this sample (N=67) in the H&Y during the ON state from before (2 [2,3]); 2.5 ± 0.6) to after (2 [2,3]); 2.2 ± 0.7) LECIG (p=0.649). However, significant changes were detected in M, N and D scores and also in the MNCD total score from pre to post-LECIG. More information is collected with the MNCD and it is also more sensible to change that the H&Y. A significant (p=0.005) decrease in the UPDRS-III during the ON-state was detected from V0 (20.9 ± 11.2) to V2 (18.1 ± 11.9) and this interesting aspect, to improve the quality of the ON state, was discussed in the original paper (Santos García et al. Eur J Neurol 2024 [epub ahead of print]). Regarding your comment, we have added this data in the manuscript and indicated this point as a limitation.

  1. I noticed that almost 50% of participants in this study have had previous device-aided therapies. Do the authors feel this may have influenced their outcomes? 

Many thanks for your comment. In patients who switched from LCIG to LECIG (34.3% for this sample) a significant reduction in the OFF time was detected as well (from 4.5 ± 2.9 to 2.3 ± 2.3; p<0.0001). If we analyze the change in the MNCD total score by groups, patients starting directly with LECIG presented a significant reduction in the score (N=44; p<0.0001). In those switching from LCIG to LECIG, a trend of significant reduction was detected despite to be a minor sample (N=23; p=0.078). Therefore, and this is something discussed in the original paper of the LECIPARK study (reference 11), LECIG could produce an improvement of symptoms in patients who have received other DAT without optimal control. This is also reflected in the MNCD.

  1. At what time point did visits take place? Was, for example, V2 a standard point in time or was this based on when a participant was seen in clinic and, therefore, possibly random? Understanding the timelines of the visits would be crucial. 

Many thanks for your comment. The V2 visit was conducted when participants were seen in clinical under daily clinical practice condition (during a period from 6 months). In the results, we explain the time of exposure to LECIG: “The mean exposure to LECIG was 172.9 ± 105.2 days (range, 7-476). Follow-up time was ≥ 3 months and ≥ 6 months in 76.2% and 47.6% of the patients, respectively”.

  1. The text in many of the figures is small and difficult to read. 

Many thanks. We have improved this aspect in the figures.

Reviewer 2 Report

Comments and Suggestions for Authors

First of all, novel approaches in PD diagnostics are welcomed. This must be done accurately to be further accepted by the community of neurologists and caregivers.

My first concern – categorization of stages by MNCD scale. Stage 1 (patient has no relevant symptoms). If none of PD symptoms is present, then is there PD at all? How can it be? This thesis needs clarification. What means “relevant”? Otherwise, stage 1 refers to a healthy (or nonPD) man.

Further staging shifts from the motor symptoms (Stage 2) to MCI (Stage 3), then to ADL (Stage 4) and ultimately to dementia (Stage 5). It seems that each stage is characterized by heterogeneous symptoms. Please put some more background behind that classification.

Then, the scoring looks simplified.  Only presence or absence of, for example, dyskinesia is considered as 1 or 0. It can be well so, that such simplification has reason and it still reflects the objective status of a PD individual. Nonetheless, this must be evidenced by comparison with conventional scoring.

Finally, the best way to compare the diagnostic strength of MNCD is, to my mind, to compare the outcome with presently existing and widely used scales, e.g. UPDRS (parts 2 and 3), SCOPA-AUT, MoCA, FAB, etc. As for the staging, H&Y probably must be also presented to compare the outcome with MNCD stages. If possible, please present data on conventional staging and scoring.

Using VAS to assess the subjective state of people with PD looks prospective.

Minor comments:

Table 1

What is DA in the table? Is it for dopamine agonist (or antagonist)?

Figure 2

It is difficult to follow coloring of axes (yellow, reddish, greenish light and dark). V2 and V0 on the right panel look identical by color, though they belong to different axes. On the left panel points are black, though there is no corresponding color on axes below (only gray is present). Please, consider better presentation and caption.

Altogether, if possible, it would be helpful if the data on traditional (conventional) scoring and staging will be provided in parallel.

Author Response

First of all, novel approaches in PD diagnostics are welcomed. This must be done accurately to be further accepted by the community of neurologists and caregivers.

My first concern – categorization of stages by MNCD scale. Stage 1 (patient has no relevant symptoms). If none of PD symptoms is present, then is there PD at all? How can it be? This thesis needs clarification. What means “relevant”? Otherwise, stage 1 refers to a healthy (or nonPD) man.

Many thanks for your comment. The description of this new tool to classify PD has been published previously in Diagnostics (Basel) (reference 4). Of course, signs/symptoms are necessary to diagnose PD but they can be minor and no impacting on the patient (mild tremor, rigidity, etc.). To decide if a symptom is relevant is according to the criteria of the neurologist based on the anamnesis, exploration, perception of the patient, influence of the symptoms over autonomy for activities, quality of life, etc. Please, see example 3 of the publication in Diagnostics (Basel) (reference 4). Two patients were classified as in MNCD stage 1 at V2 because they had at that moment an excellent response to LECIG with perfect control of symptoms without relevant impact according to the neurologist criteria. We know from experience that it is possible and can be achieved, and it is here that the impact of a DAT is truly positive for the patient and their family. These two patients had been classified at V0 in the stage 2 (with normal cognition and independence for ADL).

Further staging shifts from the motor symptoms (Stage 2) to MCI (Stage 3), then to ADL (Stage 4) and ultimately to dementia (Stage 5). It seems that each stage is characterized by heterogeneous symptoms. Please put some more background behind that classification.

Many thanks for your comment. The MNCD classification has been proposed as a tool for use in clinical practice to help neurologists have a complete and global view of the disease. It was defined by consensus of a group of experts from Spain. The stages have been proposed based on symptoms or complications that appear progressively throughout the disease and generate an impact on the patient and their family. It is not the objective of this paper to discuss the MNCD. Four papers have been published, including one of a Chinese cohort. If you want more information about the MNCD classification, please review the article in reference 4.

Then, the scoring looks simplified.  Only presence or absence of, for example, dyskinesia is considered as 1 or 0. It can be well so, that such simplification has reason and it still reflects the objective status of a PD individual. Nonetheless, this must be evidenced by comparison with conventional scoring.

Many thanks for your comment. The classification is not intended to be a scale that allows the exact quantification of the change in different symptoms (for example, the UDysRS). As I have explained, it is a tool to help give a global view of the patient's condition. If a patient has motor fluctuations and is treated with a DAT and they disappear with 0% of OFF time per day or the patient has very mild and discrete episodes with no impact at the discretion of the neurologist based on the data collected in the consultation, the score is 0. You are right that there may be variability. In this regard, a study to analyze inter and intra observer variability is pending in our country.

Finally, the best way to compare the diagnostic strength of MNCD is, to my mind, to compare the outcome with presently existing and widely used scales, e.g. UPDRS (parts 2 and 3), SCOPA-AUT, MoCA, FAB, etc. As for the staging, H&Y probably must be also presented to compare the outcome with MNCD stages. If possible, please present data on conventional staging and scoring.

The aim of this study was to describe in this cohort the results of applying the MNCD before and after been treated with LECIG. But not to compare with another scale. As we commented in the Introduction, a group from China observed in a cohort of 357 PD patients that the correlation of the MNCD staging with the QoL was more statistically significant than to the Hoehn&Yahr staging [reference 9]. The UPDRS-III and H&Y were assessed during the OFF and during the ON state before but only during the ON state after receiving LECIG (patients go to the clinic in daily clinical practice in the ON state when they are receiving levodopa infusion). No significant changes were detected in this sample (N=67) in the H&Y during the ON state from before (2 [2,3]); 2.5 ± 0.6) to after (2 [2,3]); 2.2 ± 0.7) LECIG (p=0.649). However, significant changes were detected in M, N and D scores and also in the MNCD total score from pre to post-LECIG. More information is collected with the MNCD and it is also more sensible to change that the H&Y. A significant (p=0.005) decrease in the UPDRS-III during the ON-state was detected from V0 (20.9 ± 11.2) to V2 (18.1 ± 11.9) and this interesting aspect, to improve the quality of the ON state, was discussed in the original paper (Santos García et al. Eur J Neurol 2024 [epub ahead of print], reference 11). Regarding your comment, we have added this data in the manuscript and indicated this point as a limitation.

Using VAS to assess the subjective state of people with PD looks prospective.

Many thanks for your comment. Using the VAS, the patient, caregiver, and neurologist's opinion of the change compared to before receiving treatment was collected. This was done at visit V2, with the patient present and asking him directly about his perception of the change. It is explained in Methods: “The data of visits V0 and V1 were collected from the medical records whereas the data of visit V2 were collected from a specific data report registry assessed in clinic”.

Minor comments:

Table 1

What is DA in the table? Is it for dopamine agonist (or antagonist)?

Many thanks for your comment. Yes, it is dopamine agonist. We clarify it.

Figure 2

It is difficult to follow coloring of axes (yellow, reddish, greenish light and dark). V2 and V0 on the right panel look identical by color, though they belong to different axes. On the left panel points are black, though there is no corresponding color on axes below (only gray is present). Please, consider better presentation and caption.

Altogether, if possible, it would be helpful if the data on traditional (conventional) scoring and staging will be provided in parallel.

Many thanks for your comment. We have used the colors used in the original description. For each axis a color is used, and this is how it is represented, with a lighter shade in V0 and a darker shade in V2. On the other hand, in the total score it is black and does not correspond to any axis since it is the sum of all of them. The score is represented below each bar, indicating V0 and V2. We believe that all the information is clear and a drawing with arrows indicating the axes and the reference was added below. In short, we believe that it is simple and we cannot think of a better way to represent it.

Reviewer 3 Report

Comments and Suggestions for Authors

It is true that staging tool for PD is a necessity, when we still use Hoehn-Yahr scale, developed before treatment with levodopa was introduced. Authors provide convincing evidence of usefulness of MNCD scale for monitoring patients with advance PD, treated with LECIG.

The main problem I see is that the study only looked at patients treated with LECIG, so I don't know if the results can be generalized to the entire advanced PD group, as the title suggests, or even to all patients treated with DAT. DAT (LCIG, CSAI, DBS, and the newly introduced fosdopa/foscarbidopa) have their own unique characteristics, and it is not immediately clear from the paper that MNCD would be useful in them, so the authors should make it clear to the reader that the results are limited to just one DAT. A comparison with other DATs would be interesting, but I understand that at this stage it is not feasible.

Nevertheless, I think the data presented will be of interest to people involved in the treatment of advanced Parkinson's disease.

Author Response

It is true that staging tool for PD is a necessity, when we still use Hoehn-Yahr scale, developed before treatment with levodopa was introduced. Authors provide convincing evidence of usefulness of MNCD scale for monitoring patients with advance PD, treated with LECIG.

The main problem I see is that the study only looked at patients treated with LECIG, so I don't know if the results can be generalized to the entire advanced PD group, as the title suggests, or even to all patients treated with DAT. DAT (LCIG, CSAI, DBS, and the newly introduced fosdopa/foscarbidopa) have their own unique characteristics, and it is not immediately clear from the paper that MNCD would be useful in them, so the authors should make it clear to the reader that the results are limited to just one DAT. A comparison with other DATs would be interesting, but I understand that at this stage it is not feasible.

Many thanks for your comment. We agree with you. The title has changed to “Use of the MNCD Classification to Monitor Clinical Stage and Response to Levodopa-entacapone-carbidopa Intestinal Gel Infusion in Advanced Parkinson´s Disease”.  This point has been included ad a limitation of the study: “Secondly, only patients treated with LECIG were included in this study but no with other DATs”.

Nevertheless, I think the data presented will be of interest to people involved in the treatment of advanced Parkinson's disease.

Many thanks. We really appreciate your opinion.

Round 2

Reviewer 1 Report

Comments and Suggestions for Authors

I would like to thank the authors for taking the time to reply to the comments and updating the manuscript. I feel the concerns have been adequately addressed and the manuscript improved in addition to better addressing the limitations. I have no further comments. 

Reviewer 2 Report

Comments and Suggestions for Authors

My comments are addressed